# Sustainability of facilities built under the Community-Led Total Sanitation (CLTS) implementation: Moving from basic to safe facilities on the sanitation ladder

**Hemez Ange Aurélien Kouassi**●*, **Harinaivo Anderson Andrianisa, Seyram Kossi Sossou, Maïmouna Bologo Traoré, Rikyelle Momo Nguematio**

Laboratoire Eaux, Hydro-Systèmes et Agriculture (LEHSA), Institut International d'Ingénierie de l'Eau et de l'Environnement (2iE), Ouagadougou, Burkina Faso

* aurelien.kouassi@2ie-edu.org, hemezange@gmail.com

**Editor:** D. Daniel, Gadjah Mada University Faculty of Medicine, Public Health, and Nursing: Universitas Gadjah Mada Fakultas Kedokteran Kesehatan Masyarakat dan Keperawatan, INDONESIA

## Abstract

In the context of monitoring progress towards SDG target 6.2, a household is counted to have access to sanitation if it uses at least basic sanitation services. Several approaches have been employed to help rural communities to climb up the sanitation ladder such as Community-led Total Sanitation (CLTS), whose primary target is to end open defecation through behavior change. CLTS does not subsidize sanitation facilities, but let households build their own facilities. The types and sustainability of facilities when construction is entrusted to households without guidelines remain understudied. The contribution of CLTS in achieving SDG6.2 also have not been studied. This paper addresses these gaps. Conducted in the province of Sissili in Burkina Faso, our study involved interviewing CLTS implementers, government officials, and community stakeholders. Coupled with household surveys, the data was analyzed using SPSS and Excel software. Findings indicate that CLTS succeeded in motivating households to build latrines hence escalating latrine coverage from 29.51% in 2016 (pre-CLTS) to 90.44% in 2020 (post-CLTS) in the province. However, 97.53% of latrines built were unimproved pit latrines with superstructures and without/with wooden or clay slabs and no roof, of which 19.76% collapsed during the rainy season. During this period, sanitation access rate rose from 11.9% to 17.00%. The study has therefore revealed that CLTS significantly elevates latrine coverage, yet it does not guarantee a proportional rise in sanitation access. This discrepancy results from the type of technologies generated by CLTS, which are not considered in calculating the sanitation access rate due to their unimproved nature. Consequently, further exploration of social approaches is essential, amalgamating technical and engineering aspects. Beyond socio-economic considerations, the sustainability of CLTS and the achievement of access to adequate and safe sanitation also rely on the robustness and resilience of the implemented facilities.

**Data Availability Statement:** All relevant data are fully available without restriction. The survey forms used for data collection have been made available in supplementary material. If a reader wants further data, it will be made available upon request by writing to the corresponding author: aurelien. kouassi@2ie-edu.org / hemezange@gmail.com.

**Funding:** This research was supported by the World Bank through the African Center of Excellence Project-Impact (ACE-Impact), [Grant numbers: IDA 6388-BF/D443-BF]. Mr. Hemez Ange Aurélien Kouassi, PhD student at the International Institute for Water and Environmental Engineering (2iE Burkina) is the beneficiary of this funding. The funders had no role in study design, data collection and analysis, decision to publish, or preparation of the manuscript.

**Competing interests:** The authors have declared that no competing interests exist.

# Introduction

Improved sanitation facility (usually a latrine) is "one that hygienically separates human excreta from human contact" [1]. Access to improved sanitation (latrines) has implications for many health-related outcomes among children in low and middle-income countries (LMIC), including diarrheal disease, soil-transmitted helminths, undernutrition and stunted growth [2–8]. Safe and well, managed sanitation is a goal for many localities, countries and even the United Nations [9,10]. The first step in achieving safely managed sanitation is to install and regularly use latrines. The latrines collect all the faecal matter produced by a household and contain a connected storage tank (a simple pit or a septic tank) or a sewer pipe to prevent it from entering the environment untreated [11]. Unfortunately, in low-income countries, the proportion of people using a safely managed sanitation system is only 10% [12–14]. As a result, diseases such as diarrhoea, cholera, typhoid, worm infections and stunted growth are common among children and vulnerable groups, particularly those living in rural areas and slums are the most affected [12,15].

Global consensus on the importance of sanitation has led to the inclusion of "access to adequate and equitable sanitation and the eradication of open defecation (OD)" in the Sustainable Development Goals (SDG6.2) [9]. However, more than half the world's population (around 4.2 billion people) still uses sanitation facilities that do not treat faeces. About 494 million people practice OD, 92% live in rural areas of Central and South Asia and Sub-Saharan Africa, threatening environmental sustainability and human health [16]. The water, sanitation and hygiene (WASH) sector is working to alleviate the challenges of inadequate sanitation worldwide, by investing in technologies and initiatives to promote sanitation [14]. To address these challenges and also achieve SDG6.2, several subsidized latrines construction projects have been implemented. Unfortunately, latrine subsidies have not always translated into increased latrine coverage [10]. Although apparently necessary to enable poor households to purchase latrines, latrine subsidies and donated materials did not increase latrine coverage in one study [17]. A meta-analysis has documented modest impacts of subsidized sanitation interventions on latrine coverage, access and use [18]. However, high latrine coverage has been difficult to achieve in many contexts, particularly in rural areas where materials and services are generally harder to access and more expensive [19]. One of the main obstacles to increasing sanitation coverage is the cost of providing it. According to the World Bank, it would cost around $19.5 billion a year to meet nationally-defined WASH targets [20], an amount that many of LMIC's water and sanitation ministries do not have.

Thus, to increase latrine coverage in rural areas, the once predominant focus on the provision of sanitation infrastructure (particularly toilet construction) has increasingly been complemented and/or replaced by approaches aimed at creating demand and facilitating sanitation behavior change [21–23]. The Community-Led Total Sanitation (CLTS) approach, discussed in this article, has recently become a leading approach to changing sanitation behavior and increasing latrine coverage. CLTS represents an attempt to replace earlier top-down interventions based on the provision of subsidized sanitation facilities. It also moves away from approaches aimed at changing sanitation behavior through one-way health risk education [21]. Currently, CLTS is one of the most widely deployed behavioral hygiene and sanitation interventions [24]. 36 countries have adopted CLTS as part of their national rural sanitation strategy and/or policy [25].

Although, the CLTS is effective in the short-term for eradicating OD, the long-term results are not encouraging: Open-Defecation-Free (ODF) communities revert to OD or partially use latrines [26]. Among the factors explaining these various delays in OD, there is Achilles heel of CLTS: the low quality and durability of the technologies (latrines) adopted by communities

[21,26–32]. The toilets built as part of CLTS interventions are constructed by community members using locally available materials, and their structure is often not durable, contributing to the slide towards open defecation [33,34]. Pit and superstructure collapse is common [35,36], sometimes affecting up to 40–50% of latrines [32,37]. In addition, some critics of CLTS have raised concerns about the use of unethical methods for latrine construction [15,35,38–42]. In the CLTS approach, less attention has been paid to sanitation technologies, leading to the use of latrines that are unhygienic and unsafe for users [27]. This has led to widespread promotion of poor-quality pit latrines in low-income contexts. Although this a low-quality facility does not guarantee safe separation of faeces from human contact, the pit latrine is considered the sanitation technology of choice in low-income contexts [27,43–45]. In 2013, 1.8 billion people, mainly in low-income countries, used pit latrines [46]. Efforts to improve sanitation in low-income countries over the past decade have led to an overall increase in pit latrines, but sanitation access rates still remain low [45]. Resources invested in sanitation interventions can be wasted if the basic hygiene standards of latrines are not assured, and if the technologies adopted are not sustainable [27].

Like other low and middle-income countries with high open defecation rates, low latrine coverage, low rates of improved sanitation facilities, Burkina Faso has also adopted since 2014 the CLTS approach as a strategy in rural areas to improve sanitation [47]. More than six (6) years after its adoption in Burkina Faso, the impacts of CLTS on latrine coverage, the rate of access to sanitation, the type of technologies and their sustainability are still poorly understood. While several studies around the world have demonstrated the effectiveness of CLTS in eradicating open defecation, few [10,27,28,30,32] have looked at the impact of CLTS interventions on latrine coverage, and even more on the rate of access to sanitation in the localities where it is implemented, the types of structures built and their resistance to wind and rain. Furthermore, studies have not considered whether the approach of letting households build their own latrines without guidance through their own knowledge, tools and means is the most appropriate solution.

In this paper, we address these identified gaps based on the case of CLTS implementation in the Sissili province of Burkina Faso. The main objectives of this study are as follows:

1. To examine how CLTS has or has not changed latrine coverage in Sissili Province.

2. To identify the types of technologies (latrines) that have been built as part of CLTS implementation in Sissili Province, and the reasons for their choice.

3. To investigate the collapse rate of latrines installed under the CLTS program in Sissili Province?

4. To analyze the real contribution of CLTS technologies to increasing the rate of access to sanitation in areas where the approach has been implemented.

## Materials and methods

### Study area

Burkina Faso is divided into regions, regions into provinces, provinces into municipalities and municipalities into villages. This study was conducted in Sissili, a province in west-central region of Burkina Faso, located about 145 km from Ouagadougou, the capital city of Burkina Faso. Sissili (**Fig 1**), our study area, is one of the 45 provinces of Burkina Faso. It is divided into 7 municipalities: Biéha (with 22 villages), Boura (with 25 villages), Nébiélianayou (with 11 villages), Niabouri (with 17 villages), Silly (with 31 villages), Tô (with 26 villages) and Léo (with 24 villages). The latter being the capital of the province. The province has 156 villages

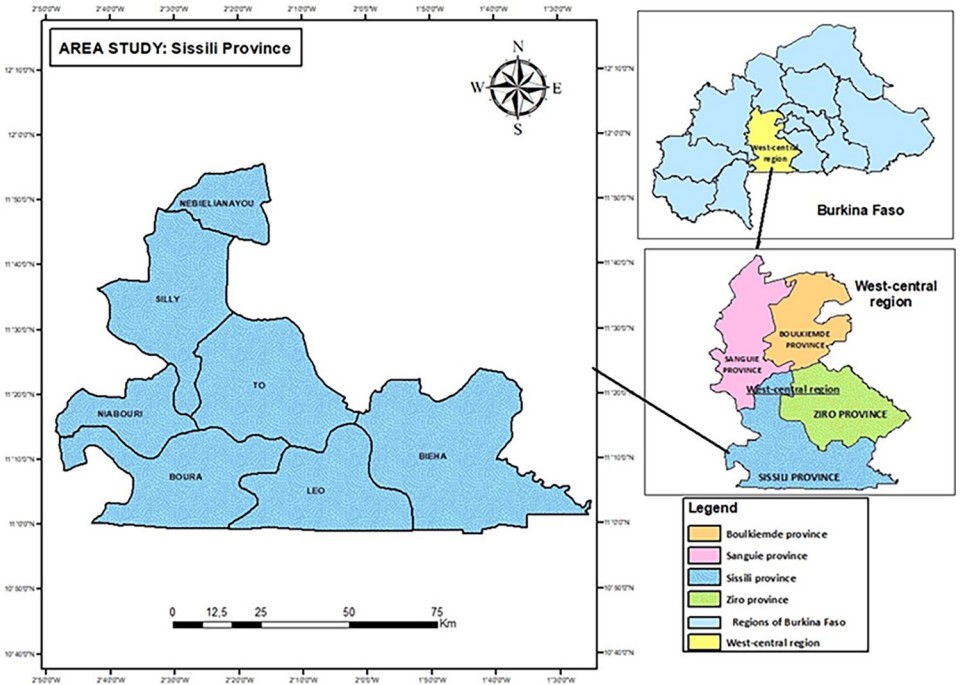

**Fig 1. Sissili province, Burkina Faso.**

and 57,703 households with a population of 337,078 people, of which 163,452 (48.49%) are men and 173,626 (51.51%) are women. It is largely rural, with only 51,743 or 15.35% of the population living in urban areas of Léo [48]. Sissili's main economic activities include small-scale agriculture, fishing, and trade. The study area was chosen using a purposive selection technique. This choice was motivated by several factors listed below:

- Sissili is the province with the highest number of ODF villages in Burkina Faso;

- The CLTS experiment (from 2016 to 2020) has been a real success here, as all 156 villages in the province have been certified ODF, making Sissili the first and only province to be declared ODF in Burkina Faso to date, whereas in the past it was a province plagued by endemic OD [49];

- As it also boasts the oldest ODF-certified villages in the country, it would be an ideal area to study the sustainability of the technologies adopted in the CLTS approach and their contribution to the province's sanitation access rate;

- The implementation of CLTS in the rural communities of Sissili was facilitated solely by the NGO Association for Peace and Solidarity (APS). It was therefore easier to collect information on the baseline situation (before CLTS) of the various villages from the aforementioned NGO;

- Access to some villages is easy and a large part of the province is not affected by terrorism, unlike most of the country.

## Methodology

The first step (1) of this study consisted in identifying the NGO having implemented the CLTS in the study area. Indeed, any NGO or structure that has to implement CLTS in a village first

establishes a baseline situation to know the number of latrines existing before the project, the number of households and the size of the village, and upon reaching ODF status also counts latrines at the end of the project. Thus, basic data on the number of latrines before CLTS and after ODF status was achieved in the province were collected from NGO APS. To ensure the reliability of these data, the same information was sought from the Ministry in charge of water and sanitation and the National Institute of Statistics and Demography (INSD). A triangulation of the data collected from these three structures was then carried out for confirmation.

In Burkina Faso, "latrine coverage" regulations stipulate that a latrine can only be used for a maximum of 10 people. Thus, the number of latrines required for full coverage (i.e. 100%) in Sissili in year n ($NLRFC_n$) was calculated using the following formula:

$$NLRFC_n = \frac{\textbf{Population size of province in year n}}{\textbf{10}}$$

As for Latrine Coverage Rates in year n ($CLR_n$) in Sissili (before and after CLTS), they were calculated using the following formula:

$$CLR_n(\%) = \frac{\textbf{Number of latrines in the province in year n}}{\textbf{NLRFC}_n} \textbf{X 100}$$

The second step (2) of this study involved household surveys in ODF-certified villages in the Sissili region. The information sought in these surveys related to the socio-economic characteristics of the household (household size, professional activities or source of income, estimated annual income) and to household sanitation (whether or not the household owned a latrine, the conditions under which the latrine was built (if the household had one), the date the latrine was built (before or after CLTS), any problems associated with the latrine, the frequency of latrine collapses and the periods during which these collapses occurred (if any)). During the survey, respondents were asked for permission to inspect the latrine. Once permission was granted, a series of observations were made. An observation grid including the type of latrine, the material used for the superstructure, the roof, the door, the material used to make the slab and whether or not there was a vent pipe was drawn up for this purpose. Following the inspection, the reasons for the type and technologies adopted by the households were collected.

The sample size of households to be surveyed "**N**" was calculated using the formula of [50] and [51]:

$$N = \frac{\textbf{t}_p^2 * \textbf{P} * (1 - \textbf{P}) * (1 + \textbf{t}_{nr})}{\textbf{y}^2}$$

with:

**P**: expected proportion of a population response or actual proportion. In the field of sanitation, it is set to 0.5 by default, which allows to have the largest possible sample [52]. In the present study, we have retained an expected proportion of a response of 0.5.

**y**: Margin of sampling error. The margin of error represents the range of certainty within which the responses obtained are accurate. It is usually between 1 and 10%. In this study, we have retained a margin of error of 5%.

**$t_{nr}$**: non-response rate. The non-response rate is considered acceptable when it is less than 10% [53]. In this study, we used a non-response rate of 5%.

**$t_p$**: sampling confidence level. In this study, we have chosen a 95% confidence level. $t_p$ represents the Z-score derived from the desired confidence level. For a confidence level of 95%, the corresponding $t_p$ value is 1.96.

This formula was chosen for two reasons: Firstly, it maximizes the sample size, since it includes the non-response rate. As a result, it also increases the reliability of the results obtained. Secondly, it is one of the most widely used formulas for sanitation studies. Several authors, such as [54–57] used it to determine the sample size for health, sanitation and epidemiological surveys.

The calculated sample size was N = 404 households to be surveyed in the province. But 410 households were actually surveyed in Sissili for this study.

Once the size of the households to be surveyed had been determined, a two-stage sampling method was adopted. The first involved identifying the villages where the surveys would take place. Purposive sampling was used to select the villages. Based on the number of municipalities (7) in the province of Sissili, seven villages (Koalga, Nadion, Boutiourou, Kayero, Onliasson, Don and Fido) were selected. These villages were selected on the basis of two criteria: the age of their ODF status (certified ODF for at least 36 months) and their easy accessibility by car from the city of Léo, the capital of Sissili Province (due to security issues related to terrorism). ODF certification dates were collected from the ministry in charge of water and sanitation. The second phase involved selecting the households to be surveyed in the villages previously selected. Systematic random sampling was then used to select households from the villages. Within the communities, household selection followed the random route method [58]. The calculated sample size **N** was distributed in proportion to the total number of households in each village. On this basis, the number of households to be surveyed in each selected village was determined.

Respondents had to be over 18 years old to answer questions. Household surveys were conducted in local languages (Moorée, Gourounsi and Fulfuldé) for better understanding by respondents. A voice recorder was used to record interviews when the interviewer allowed it. A camera was used to capture images of latrines during data collection. A team of 4 data collectors who understood the local languages was trained on the questionnaire before the survey. The training included the use of the KoboCollect mobile application on which the questionnaire was deployed and the characteristics of different types of latrines.

The household surveys were conducted during the rainy season in two campaigns. The first collection campaign, from August 1st to 31st, 2021, covered 4 villages and the second, from July 1st to 15th, 2022, covered the remaining 3 villages. The months of July, August and September correspond to the rainy season in Burkina Faso. The choice of this period as the data collection period was made based on the literature that revealed that many latrines collapsed due to rainfall.

The third step (3) of this study involved researching sanitation access rates in the province before and after CLTS. To do this, interviews and an analysis of annual activity reports between 2016 and 2016 from the Provincial Department of Water and Sanitation of Sissili were carried out. The quantitative data obtained during these interviews and the documents on these rates were then checked and compared with those available to the ministry in charge of water and sanitation in Burkina Faso and published on its website. This was done to ensure the reliability of the data collected.

Table 1 summarizes how the survey respondents were grouped.

- The fourth and final step of the study was data analysis. The analysis began by organizing data from recorded interviews, field notes taken during observations, household surveys, and reports of CLTS implementation in Sissili by the NGO APS, and the annual balance sheets of the INSD and the Ministry in charge of water and sanitation between 2016 and 2020. Two analysis procedures were used based on mixed-methods approach. Quantitative data analysis followed standard procedures for identifying, entering and manipulating

**Table 1. Grouping the survey respondents.**

| Stakeholders | Actors Met | Tools for Data Collection | Sampling | Number of Respondents | | |
|---|---|---|---|---|---|---|
| | | | | Men | Women | Total |
| NGO implementing | Association for Peace and Solidarity (APS) | Individual Interview Guide | Purposive selection | 8 | 4 | 12 |
| Institutional actors | Ministry of Water and Sanitation | Individual Interview Guide | Purposive selection | 0 | 2 | 2 |
| | Provincial Department of Water and Sanitation of Sissili | Individual Interview Guide | Purposive selection | 2 | 0 | 2 |
| | National Institute of Statistics and Demography (INSD) | Individual Interview Guide | Purposive selection | 1 | 1 | 2 |
| Community stakeholders | Traditional and religious leaders | Individual Interview Guide | Purposive selection | 9 | 0 | 9 |
| | Private operators (masons, shopkeepers) | Individual Interview Guide | Purposive selection | 5 | 0 | 5 |
| Households | Men and women | Household survey | Random selection | 252 | 158 | 410 |
| TOTAL | | | | 277 | 165 | 442 |

variables using Statistical Package for the Social Sciences (SPSS) and EXCEL. The different graphs provided by SPSS were reproduced in Excel. Descriptive statistics including frequencies, proportions for categorical variables and means for continuous variables were used to describe the study subjects at the univariate level (proportion of different types of latrines in each village, number of latrines before and after CLTS, rate of access to sanitation before and after CLTS, etc.). Analysis of the qualitative data used hand-coding procedures. First, the qualitative data were analyzed using thematic and content analysis approaches [59,60]. As recommended by [61], qualitative data analysis procedures must ensure data coding. Data coding involves a systematic examination of the text to identify certain ideas, phrases, sentences, and quotes that represent certain phenomena and show what the data represent. A list of factors explaining the type of latrines with the highest proportion under CLTS implementation was developed based on the responses provided by the respondents.

## Ethical considerations

In this study, it was necessary to collect information of personal nature. Therefore, anonymity was of vital importance to protect the personal identity of the respondents. Necessary precautions were taken to protect the confidentiality of respondents. Participants were duly informed about the purpose of the study. They were also made aware of the main components of the research design. Respondents were assured of confidentiality, even if they were told, for example, in the case of personal interviews, that their voices would be recorded. No participant was coerced by any means to take part in the study. They voluntarily agreed to actively participate in the study. Oral informed consent was obtained from all participants. The ethical rules set forth by the Research Ethics and Deontology Committee of the 2iE Institute (N˚2023/01/DG/ SG/DR/HK/fg) were respected. The research committee had given its approval for this study.

## Result

### Evolution of latrine coverage in the province of Sissili between 2016 and 2020

In 2016 (before the CLTS in the province), with an estimated population of 236,014 Sissili had 6,976 latrines, including 1,515 in the municipality of Bieha, 969 in Boura, 185 in Nébiélia-nayou, 417 in Niabouri, 430 in Silly, 1,870 in Tô and 1,590 in Léo. The number of latrines required for the province to achieve full coverage in 2016 ($NLRFC_{2016}$) was 23,602, i.e. around 16,626 latrines in addition to those already in existence. The latrine coverage rate in 2016 ($CLR_{2016}$) was 29.51%.

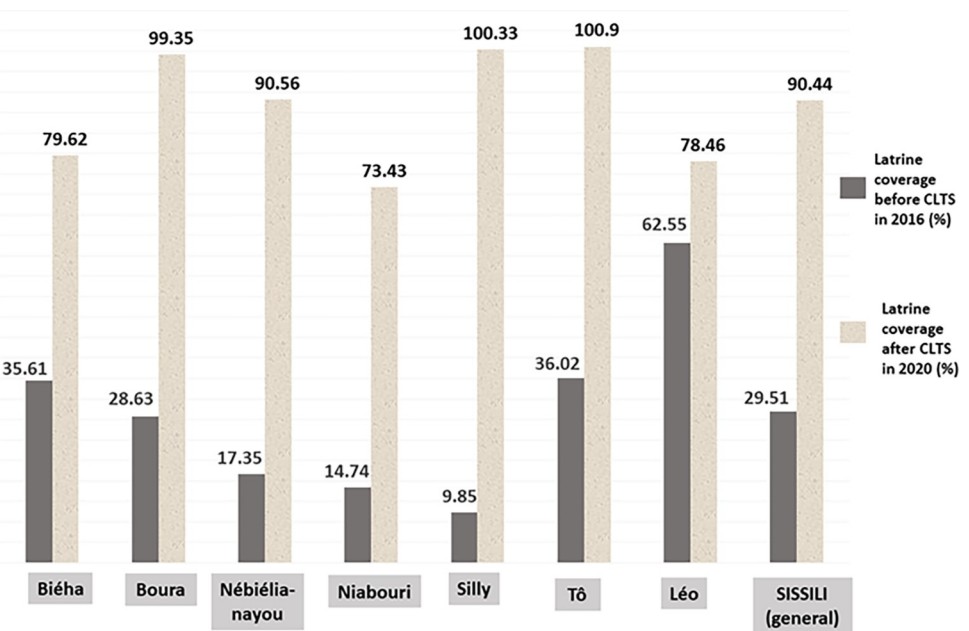

**Fig 2. Evolution of the latrine coverage rate in all the municipalities of Sissili from 2016 to 2020.**

Following the implementation of the CLTS (from 2016 to 2020), around 18,471 new latrines were built in the province, bringing the number of latrines in Sissili to 25,447 in 2020, including 4,199 in the municipality of Bieha, 4,019 in Boura, 1,190 in Nébiélianayou, 2,231 in Niabouri, 5097 in Silly, 6056 in Tô and 2,655 in Léo, for an estimated population of 281,335. The $NLRC_{2020}$ was 28134. The latrine coverage rate in 2020 ($CLR_{2020}$) was 90.44%.

The latrine coverage rate in Sissili has therefore increased significantly from 29.51% in 2016 to 90.44% in 2020 after the CLTS interventions.

**Fig 2** and **Table 2** present a summary of the data on latrine coverage in all the municipalities of Sissili in 2016 (year the CLTS triggered) and 2020 (the year the province was declared ODF).

**Table 2. Summary of latrine coverage data in all Sissili municipalities from 2016 and 2020.**

| | Sissili | | | | | | | |
|---|---|---|---|---|---|---|---|---|
| Municipalities | Biéha | Boura | Nébiélianayou | Niabouri | Silly | Tô | Léo | Total |
| Number of villages | 22 | 25 | 11 | 17 | 31 | 26 | 24 | **156** |
| Population size | 52 733 | 40 448 | 13 137 | 30 371 | 50 795 | 60 020 | 33 831 | **281 335** |
| Number of existing latrines in 2016 (before CLTS) | 1 515 | 969 | 185 | 417 | 430 | 1 870 | 1 590 | **6 976** |
| Total number of latrines required for 100% coverage in 2020 | 5 274 | 4 045 | 1 314 | 3 038 | 5 080 | 6 002 | 3 384 | **28 134** |
| Number of latrines built between 2016 and 2020 | 2 684 | 3 050 | 1 005 | 1 814 | 4 667 | 4 186 | 1 065 | **18 471** |
| Total number of latrines in 2020 | 4 199 | 4 019 | 1 190 | 2 231 | 5 097 | 6 056 | 2 655 | **25 447** |
| Latrine coverage rate in 2016 (%) | 35.61 | 28.63 | 17.35 | 14.74 | 9.85 | 36.02 | 62.55 | **29.51** |
| Latrine coverage rate in 2020 (%) | 79.62 | 99.35 | 90.56 | 73.43 | 100.33 | 100.90 | 78.46 | **90.44** |

## Evolution of latrine coverage in the villages surveyed between 2016 and 2020

Like the general trend in latrine coverage in Sissili since the introduction of the CLTS in 2016, the number of latrines has also increased in the seven villages surveyed. This increase in latrine coverage in the villages was attributed by the various respondents entirety (98%) to the CLTS, since no other project was implemented simultaneously in Sissili. Indeed, the village of Kayer-oThio went from 38 latrines in 2016 to 190 latrines in 2017 (when it obtained its ODF certification) for an estimated population of 1,837 inhabitants, i.e., a coverage rate of 103.26%; the village of Koalga from 44 latrines in 2016 to 128 latrines in 2017 for an estimated population of 1,335 inhabitants, i.e., a coverage rate of 95.52%; the village of Boutiourou from 27 latrines in 2016 to 132 latrines in 2017 for an estimated population of 1,515 inhabitants, i.e., a coverage rate of 86.84%; the village of Nadion from 29 latrines in 2016 to 230 latrines in 2018 for an estimated population of 2,269 inhabitants or a coverage rate of 101.32%; the village of Onliassan from 40 latrines in 2016 to 252 latrines in 2017 for an estimated population of 2,786 inhabitants or a coverage rate of 90.32%; the village of Don from 34 latrines in 2016 to 224 latrines in 2018 for an estimated population of 2,163 inhabitants i.e. a coverage rate of 103.23%; the village of Fido from 14 latrines in 2016 to 121 latrines in 2017 for an estimated population of 1,372 inhabitants i.e. a coverage rate of 87.68%. The results of the household surveys revealed that 68.55% (255/372) of the respondents who had latrines during the data collection period did not have latrines before the implementation of CLTS in their village. Specifically, this was 52.94% (27/51) of respondents in KayeroThio village, 48.98% (24/50) in Koalga village, 68.97% (40/58) in Boutiourou, 69.09% (38/55) in Nadion, 69.05% (29/42) in Onliassan, 77.42% (48/62) in Don and 89.09% (49/55) in Fido.

Households that did not have latrines said they had latrines 3 to 4 years earlier, especially on the day of the village ODF-status assessment. However, these latrines had collapsed several times following rainfall for some and had full pits for others. To maintain the solidarity of the neighborhood, the villagers whose toilets had collapsed used the nearest neighbor's latrine to defecate.

Table 3 presents a summary of the data in terms of latrine coverage in the villages surveyed.

## Type of latrines built during the implementation of the CLTS in Sissili

On the 410 households surveyed, 372 (90.73%) had a latrine. The inspection of these latrines revealed three (3) types: Unimproved pit latrines (Fig 3), SanPlat latrines (Sanitary Platforms) (Fig 4) and VIP latrines (Ventilated Improved Pit) (Fig 5).

Pit latrines are the simplest form of dry latrine. A pit latrine usually consists of three main parts: a hole in the ground, a slab or floor with a small hole, and a shelter. The pit is usually at least two meters deep and one meter wide (Fig 3A). They should also have a cover that can be placed over the hole to reduce fly and odor problems. Pit latrines should have also a masonry upper part, called a superstructure, which protects from rain and sun and provides privacy and comfort for the user. However, it was found that the characteristics of the traditional or simple pit latrines encountered in Sissili were different from those in the literature. The majority of pit latrine slabs built in Sissili were made of wood (Fig 3D), flooring or clay. Few pit latrines had a concrete defecation slab. The other feature was that all (100%) of the pit latrines inspected were without roofs (Fig 3B and 3C), thus not protecting the slabs in case of rain. In addition, the superstructures were made of terracotta or clay (Fig 3B–3D).

The characteristics of the SanPlat latrines (Fig 4) and VIP latrines (Fig 5) found in Sissili were similar to those defined in the literature.

**Table 3. Summary of the data in terms of latrine coverage in the villages surveyed.**

| | | KayeroThio | Koalga | Boutiourou | Nadion | Onliassan | Don | Fido | Total |
|---|---|---|---|---|---|---|---|---|---|
| CLTS trigger date | | March 2016 | April 2016 | March 2016 | May 2016 | April 2016 | April 2016 | June 2016 | |
| Date ODF-status was achieved | | June 2017 | June 2017 | Sept 2017 | July 2018 | Oct 2017 | Oct 2018 | June 2017 | |
| Population size | | 1 837 | 1 335 | 1 515 | 2 269 | 2 786 | 2 163 | 1 372 | **13 277** |
| Total number of latrines required for 100% coverage | | 184 | 134 | 152 | 227 | 279 | 217 | 138 | **1 331** |
| Number of households | | 309 | 245 | 310 | 416 | 550 | 296 | 250 | **2 376** |
| Number of latrines before CLTS | | 38 | 44 | 27 | 29 | 40 | 34 | 14 | **206** |
| Latrine coverage rate before CLTS (%) | | 20.65 | 32.84 | 17.74 | 12.78 | 14.34 | 15.67 | 10.14 | **16.98** |
| Number of latrines on the day of ODF certification | | 190 | 128 | 132 | 230 | 252 | 224 | 121 | **1 277** |
| Later coverage rate after CLTS (%) | | 103.26 | 95.52 | 86.84 | 101.32 | 90.32 | 103.23 | 87.68 | **95.94** |
| Number of households surveyed | | 60 | 50 | 61 | 62 | 60 | 62 | 55 | **410** |
| Latrine in surveyed households | Households with at least one latrine | 51 | 49 | 58 | 55 | 42 | 62 | 55 | **372** |
| | Households with at least one latrine (%) | 85.00 | 98.00 | 95.08 | 88.71 | 70.00 | 100.00 | 100.00 | **90.73** |
| | Households without latrine | 9 | 1 | 3 | 7 | 18 | 0 | 0 | **38** |
| | Households without latrine (%) | 15.00 | 2.00 | 4.92 | 11.29 | 30.00 | 0.00 | 0.00 | **9.27** |
| Date of construction of latrines in households surveyed | After CLTS | 27 | 24 | 40 | 38 | 29 | 48 | 49 | **255** |
| | After CLTS (%) | 52.94 | 48.98 | 68.97 | 69.09 | 69.05 | 77.42 | 89.09 | **68.55** |
| | Before CLTS | 24 | 25 | 18 | 17 | 13 | 14 | 6 | **117** |
| | Before CLTS (%) | 47.06 | 51.02 | 31.03 | 30.91 | 30.95 | 22.58 | 10.91 | **31.45** |

Unimproved pit latrines constituted the majority of latrines built by households (87.63%; 326/372), followed by VIP latrines (9.14%; 34/372), while SanPlat latrines (3.23%; 12/372) came last. No flush or EcoSan latrines were found in the villages surveyed. More than half (65.21%; 30/46) of the households that had VIP and SanPlat latrines said they had acquired them before the implementation of the CLTS in their village. They were acquired through previous latrine projects in their village, some of which subsidized between 50% and 75% (slabs, doors, roof, ventilation pipe, etc.) of the latrine for households that wanted one, and others that donated the latrine outright. The 73.31% of the pit latrines were built under the CLTS. The choice of this type of latrine was motivated by several reasons according to the respondents, the main ones being: the low cost of construction (ranging from 30 to 82 USD) (cited by 218 respondents), no need of construction technical expertise (cited by 144 respondents), and it can be built with any locally available material (not necessarily cement) (cited by 77 respondents), requires less water for its construction than approved latrines (VIP, EcoSan, flush toilets, etc.) (cited by 54 respondents) and it is the most widespread in the village (cited by 36 respondents).

Table 4 presents the types of latrines in the surveyed villages and their dates of construction.

## Collapses and sustainability of latrines built under the CLTS

The 38 (9.27%) households surveyed who did not have latrines claimed to have had them 3 to 4 years earlier, especially on the day of the village's ODF status check. However, three main reasons were given to explain the absence of latrines: the frequent collapse of latrines (during the rainy season), which discouraged them because it required constant financial, material and

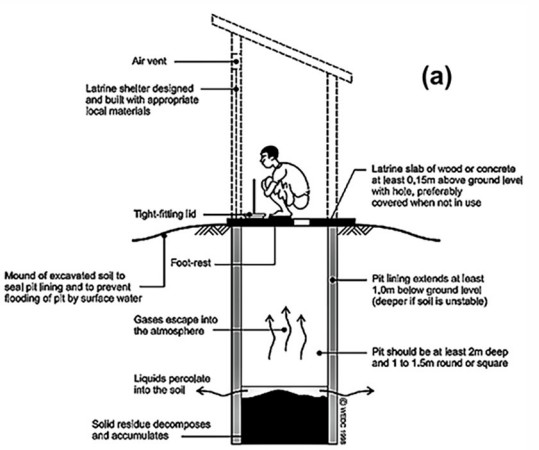
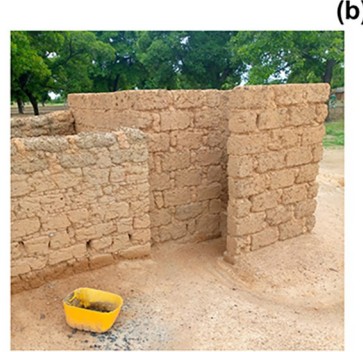
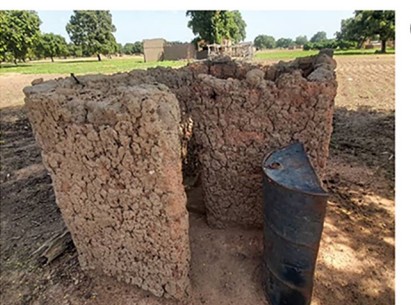
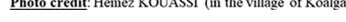
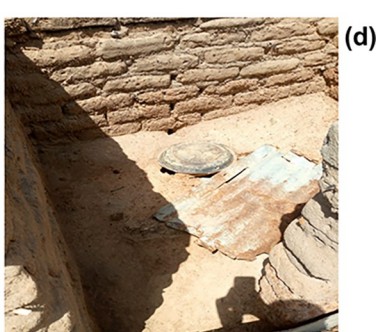

**Fig 3. Schematic and photos of a pit latrines.**

human investment; and the pit was full and therefore unusable, so the defecation hole was closed and the latrine was used for other purposes (chicken breeding or storage). Latrines collapsed in each of the seven villages according to testimonies and surveys. These collapses occurred during the previous rainy seasons. The majority (97.53%) were unimproved pit latrines. This is understandable given that the superstructures of these latrines are mostly made of wood, earth, and straw, materials that cannot withstand heavy rains and end up rotting or being damaged by termites.

Two people interviewed during the surveys in the village of Onliassan said they were aware that the latrine construction materials they used were not of good quality and could not withstand the rains and strong winds. But given their financial situation, they have no choice.

"... To build our latrines, we use local materials that are not very resistant to heavy rains. They can't withstand it because we used clay and dry wood. Since they cannot resist, the latrines collapse every rainy season. We build them ourselves. We do our best. Unfortunately, we don't have enough money to build them with cement...".—interview, male, Onliassan village.

"... My neighbor is an elderly woman. Her wooden latrine collapsed twice. She got tired of it and gave up. In addition, in our village, the soil is hard and it is difficult to dig every time for latrine construction...".—interview, male, Onliassan village.

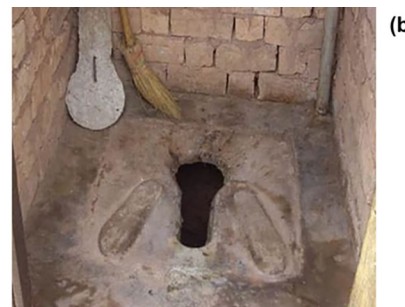

Photo credit: Hemez KOUASSI (in the village of Koalga)

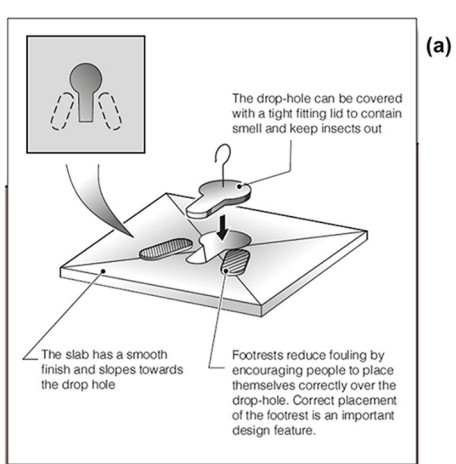

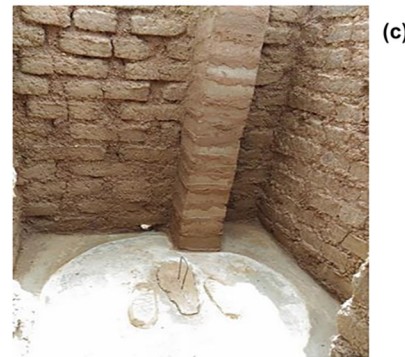

Photo credit: Hemez KOUASSI (in the village of Don)

**Fig 4. Schematic and photos of a SanPlat latrines.**

It should also be noted that only 29 households (7.07%) said they had used the expertise of a mason for the construction of their latrine, because this would increase the budget for the construction of the latrine. The cost of building a sustainable latrine is almost the same as (re) building cheap latrines. In the same way that owning a latrine is cheaper than paying hospital bills.

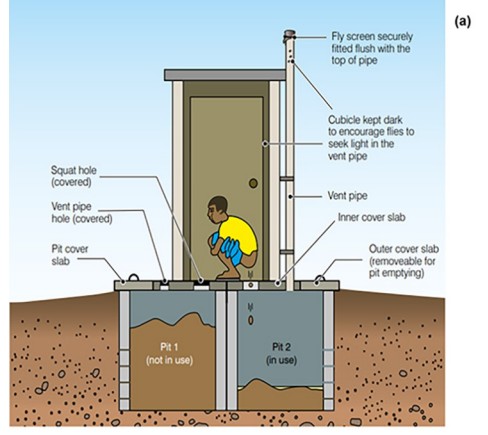

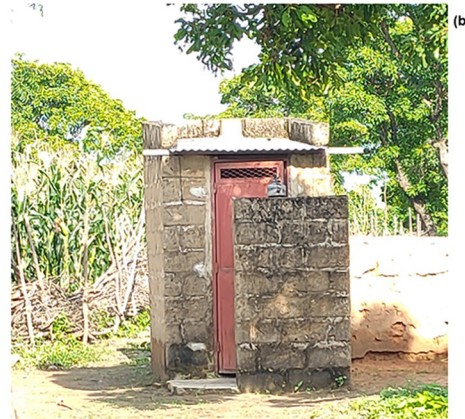

Photo credit: Hemez KOUASSI (in the village of Koalga)

**Fig 5. Schematic and photos of a Ventilated Improved Pit (VIP) latrines.**

**Table 4. Types of latrines in the surveyed villages and their dates of construction.**

| | | | KayeroThio | Koalga | Boutiourou | Nadion | Onliassan | Don | Fido | Total |
|---|---|---|---|---|---|---|---|---|---|---|
| Latrine in surveyed households | Households with latrine | | 51 | 49 | 58 | 55 | 42 | 62 | 55 | 372 |
| Type of latrine | Unimproved pit latrines | | 42 | 39 | 55 | 46 | 37 | 58 | 49 | 326 |
| | Unimproved pit latrines (%) | | 82.35 | 79.60 | 94.83 | 83.64 | 88.10 | 93.55 | 89.10 | 87.63 |
| | VIP latrines | | 7 | 6 | 3 | 8 | 5 | 1 | 4 | 34 |
| | VIP latrines (%) | | 13.72 | 12.24 | 5.17 | 14.55 | 11.90 | 1.61 | 7.27 | 9.14 |
| | Sanplat | | 2 | 4 | 0 | 1 | 0 | 3 | 2 | 12 |
| | Sanplat (%) | | 3.92 | 8.16 | 0.00 | 1.81 | 0.00 | 4.84 | 3.63 | 3.23 |
| Date of construction of latrines | Unimproved pit latrines | Before CLTS | 18 | 20 | 17 | 11 | 10 | 11 | 0 | 87 |
| | | Before CLTS (%) | 42.86 | 51.28 | 30.91 | 23.91 | 27.03 | 18.97 | 0.00 | 26.69 |
| | | After CLTS | 24 | 19 | 38 | 35 | 27 | 47 | 49 | 239 |
| | | After CLTS (%) | 57.14 | 48.72 | 69.09 | 76.09 | 72.97 | 91.03 | 100.00 | 73.31 |
| | VIP latrines | Before CLTS | 4 | 3 | 1 | 5 | 3 | 1 | 4 | 21 |
| | | Before CLTS (%) | 57.14 | 50.00 | 33.33 | 62.50 | 60.00 | 100.00 | 100.00 | 61.76 |
| | | After CLTS | 3 | 3 | 2 | 3 | 2 | 0 | 0 | 13 |
| | | After CLTS (%) | 42.86 | 50.00 | 66.67 | 37.50 | 40.00 | 0.00 | 0.00 | 38.24 |
| | Sanplat | Before CLTS | 2 | 2 | 0 | 1 | 0 | 2 | 2 | 9 |
| | | Before CLTS (%) | 100.00 | 50.00 | 0.00 | 100.00 | 0.00 | 66.67 | 100.00 | 75.00 |
| | | After CLTS | 0 | 2 | 0 | 0 | 0 | 1 | 0 | 3 |
| | | After CLTS (%) | 0.00 | 50.00 | 0.00 | 0.00 | 0.00 | 33.33 | 0.00 | 25.00 |

**Fig 6** shows some photos of collapsed latrines.

As for the 327 households that had latrines, 51 households (15.60%) reported that this was not their first latrine. Indeed, they have seen their previous latrines collapse following heavy rains or very strong winds but they have rebuilt them. **Table 5** presents the number and type of collapsed latrines in the villages surveyed after the implementation of the CLTS.

## Evolution of the sanitation access rate in Sissili before and after the CLTS

Sissili, although having gone from 6,976 latrines (with a coverage rate of 29.51%) in 2016 to 25,447 latrines (with a coverage rate of 90.44%) in 2020, saw its sanitation access rate rise from 11.9% in 2016, to 12.4% in 2017, then to 13.1% in 2018, then to 15% and to 17% in 2020 (**Fig 7**). Thwarting all predictions, which predicted one of the highest rural sanitation access rates in the country, the reality was quite different. The rate of access to sanitation in Sissili in 2020 (17%) was lower than the national rural access rate of 19.9% in 2020.

The ODF declaration of a community implies that the practice of OD is banned within the community and that every household has a latrine. But although Sissili is the only province currently declared ODF following the implementation of CLTS in Burkina Faso, its sanitation access rate increased by only 5 points between 2016 and 2020.

## Discussion

Our findings revealed that the implementation of the CLTS in Sissili has significantly increased latrine coverage in Sissili. This result agrees with several other findings from CLTS implementation in 11 countries in Africa and Asia [62,63]. However, a high level of latrine coverage, as is the case in the province of Sissili, does not in itself provide an objective assessment of access to sanitation, let alone the use of latrines in an area [54,64,65]. Indeed, some CLTS studies have found returns to open defecation 2–4 years after the end of CLTS activities [26,35,66] in communities that had high latrine coverage. One of the reasons cited among many is the type of

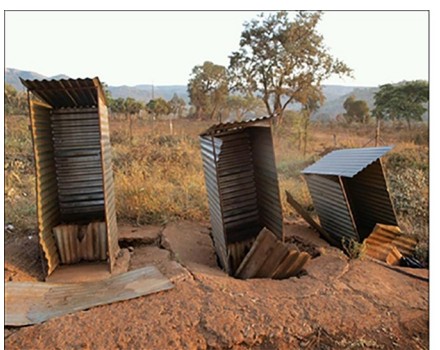

(a)

Collapsed latrine at Fido

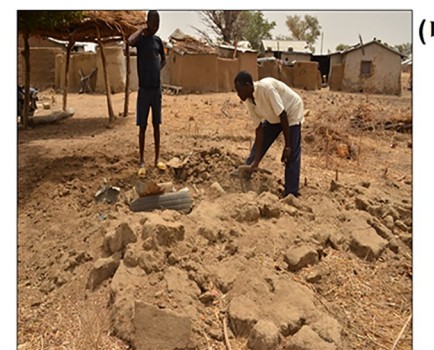

(b)

Collapsed latrine at Boutiourou

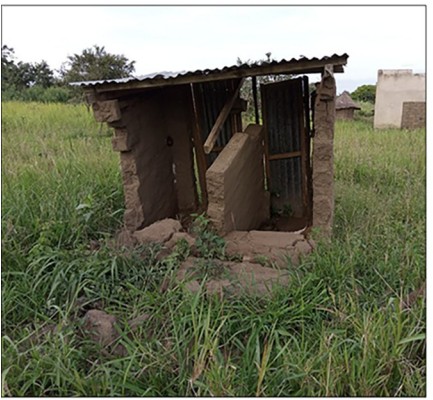

(c)

Collapsed latrine at Nadion

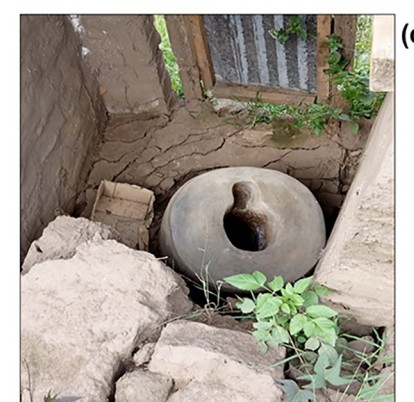

(d)

Collapsed latrine at Don

**Fig 6. Collapsed latrines.**

**Table 5. Number and type of collapsed latrines.**

| | | KayeroThio | Koalga | Boutiourou | Nadion | Onliassan | Don | Fido | Total |
|---|---|---|---|---|---|---|---|---|---|
| **Number of households surveyed** | | **60** | **50** | **61** | **62** | **60** | **62** | **55** | **410** |
| Latrine in surveyed households | Households with latrine | 51 | 49 | 58 | 55 | 42 | 62 | 55 | **372** |
| | Households without latrine | 9 | 1 | 3 | 7 | 18 | 0 | 0 | **38** |
| Households having recorded at least one latrine collapse after CLTS | Number of collapses | 15 | 8 | 11 | 13 | 17 | 7 | 10 | **81** |
| | Proportion of collapses (%) | 25.00 | 16.00 | 18.03 | 20.97 | 28.33 | 11.29 | 18.18 | **19.76** |
| Season of collapse | Rain season | 15 | 8 | 11 | 13 | 17 | 7 | 10 | **81** |
| | Rain season (%) | 100.00 | 100.00 | 100.00 | 100.00 | 100.00 | 100.00 | 100.00 | **100.00** |
| | Dry season | 0 | 0 | 0 | 0 | 0 | 0 | 0 | **0** |
| | Dry season (%) | 0.00 | 0.00 | 0.00 | 0.00 | 0.00 | 0.00 | 0.00 | **0.00** |
| Type of collapsed latrines | Unimproved pit latrine | 15 | 7 | 11 | 13 | 17 | 6 | 10 | **79** |
| | Unimproved pit latrine (%) | 100 | 87.5 | 100 | 100 | 100 | 85.71 | 100 | **97.53** |
| | VIP latrines | 0 | 0 | 0 | 0 | 0 | 0 | 0 | **0** |
| | VIP latrines (%) | 0.00 | 0.00 | 0.00 | 0.00 | 0.00 | 0.00 | 0.00 | **0.00** |
| | SanPlat | 0 | 1 | 0 | 0 | 0 | 1 | 0 | **2** |
| | SanPlat (%) | 0.00 | 12.5 | 0.00 | 0.00 | 0.00 | 14.29 | 0.00 | **2.47** |

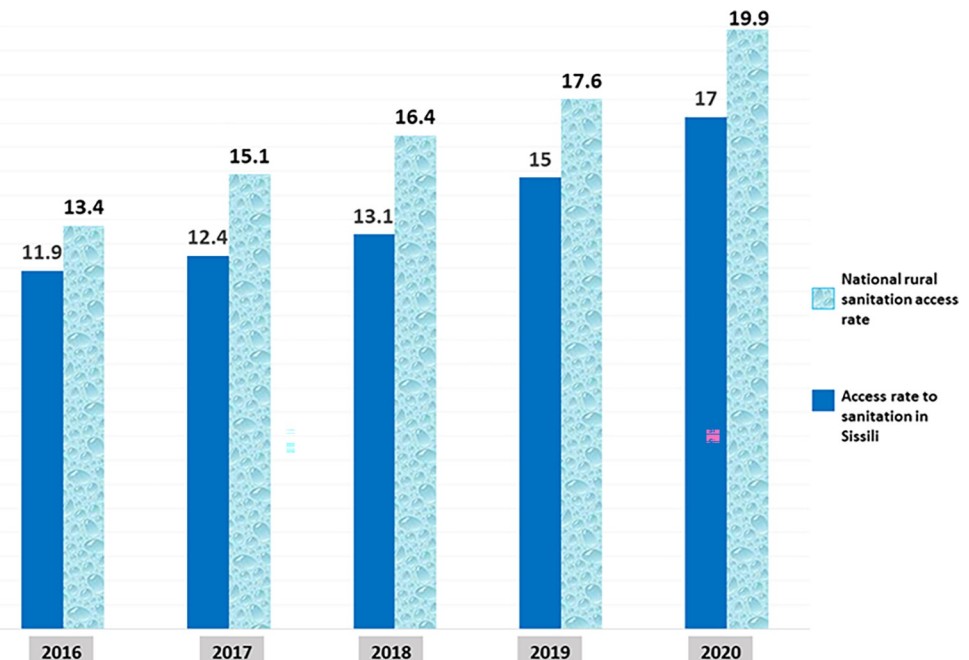

**Fig 7. Evolution of the sanitation access rate in Sissili since the implementation of the CLTS compared to the sanitation access rate in rural areas at the national level.**

latrine resulting generally from CLTS that was not strong enough to withstand the weather especially the rains. Indeed, because of the freedom to choose the type of latrine it allows, CLTS in Sissili has led to a large number of poor latrines (usually unimproved), often endangering the lives of users. Sissili is not an exception in terms of the type of latrine resulting from CLTS, as in [67], conducted in Ghana, Mali, Niger and Nigeria, 70.5% (95% CI 65.7–74.8) of the surveyed households had a pit latrine. These pit latrines, which are also mostly built in Sissili, although they help to combat OD, have collapsed in some cases. These pit latrine collapses following rains were observed in Gambia [68] where the pit had completely collapsed in 64 (79.0%) of the 81 unusable latrines and partially collapsed ("unsafe") in 13 (16.0%). Almost all pit latrine collapses identified in the literature, particularly in West Africa (Ghana, Gambia, Mali, Niger, Nigeria, and Burkina Faso), including those in Sissili, occurred during the rainy season [68,69]. Frequent disasters such as floods that result in collapsed pits and latrines have deterred ODF communities from rebuilding their latrines, as reported in the Bangladesh study [66]. In addition to the rains, [70] reported that termites attacked the slabs of some pit latrines that were made of wood beams or branches which severely weakened the latrine structure often leading to its collapse. Local pit latrines were not expected to last an entire rainy season. Improved latrines, made with a cement slab, were highly desirable because they offered a great improvement in convenience, safety, and cleanliness, but the cost of materials and the difficulties of transporting materials to villages prevented people from building them [68].

Although there is a wide range of sanitation technology options for vulnerable communities, few publications explore their applicability to CLTS [71]. Our analysis suggests that there is no single universal technology capable of solving the problem of sustainability of CLTS achievements, but rather a set of different technological arrangements that could be implemented taking into account the environmental and social contexts. Indeed, local environmental conditions can be limiting factors for the implementation of a sanitation technology,

especially in areas that are prone to natural flooding [72]. Scientific publications regarding on-site sanitation technologies published in recent years reveal a variety of options, including water-based solutions, urine-diverting toilets [73], composting toilets [74], and ecological sanitation (Ecosan) [75], all of which have great potential for application in rural communities. A sensitivity analysis using polytomous regression to assess the sustainability of different sanitation technologies revealed that flush toilets were more likely to be sustainable than other types of improved latrines and obviously than pit latrines [62]. However, in the context of global water scarcity, water-intensive flush latrines can no longer be considered the ideal target for access to safe sanitation. Indeed, in rural areas, communities are very often faced with water access problems. Global population growth and the effects of climate change will exacerbate this water access problem in these already vulnerable communities. Lack of water may also be a reason for these communities to turn to less water-intensive or non-water-intensive types of latrines such as pit latrines in the CLTS [26]. Research conducted with rural communities in South Africa and Brazil [76] showed that users find flush toilets more desirable because they associate a certain degree of social status with their use, resilience, and also find them easier to maintain and clean than dry alternatives.

The CLTS does not provide suggestions for latrine designs or materials used to construct latrines. Given the reported link between latrine durability and type of construction materials, it would have made more sense to opt for sustainable latrine designs from the start, rather than trying to move populations up on the sanitation ladder over time. Studies by [28,32,77] on latrine upgrading after achieving ODF status report no or little shifts upwards on the sanitation ladder by households. The "no-assistance" or "no-subsidy" principle, which is rarely questioned, is one of the causes of the slippage to OD in some communities where CLTS was triggered because a material chosen based on natural constraints (slope, soil type, flooding, etc.) is likely to be more sustainable than one chosen with the sole purpose of achieving ODF status. However, it is undeniable that the quality, availability and cost of a material are related. But technological aspects alone are not sufficient to ensure the sustainability of a latrine. Social aspects are considered essential and, if not adequately addressed, the failure of the implemented solution is likely to occur [78–80]. A number of community, household, and structural factors besides technological factors were found to be associated with better toilet durability, including households with higher socioeconomic status, households with disabled members, living in communities with higher basic sanitation coverage, and having toilets in an area with a shallower water table [62,70]. Several other studies have reported that frequent personal contact with health promoters and accountability over a period of time [81] an enabling environment with market access to latrine products [82–84], follow-up monitoring [17,85], social cohesion and social capital among community members [50], effective community leadership and political will [17], access to markets and sanitation materials, and sustainable behavior change [17,50] can lead to increased sustainability of latrines.

The choice of a technology to use, in addition to being sustainable, must be made following community participation as intended by the CLTS principles. Those who will be the users of the proposed sanitation solutions should be involved from the planning phase of the actions to implement the toilets. There are studies that have identified how participation in the selection process [86] optimizes people's ownership of the sanitation technique. Another important factor in determining the sustainability of a given technology is its social acceptance by the local community [68,78,87]. It is important to analyze user feedback on how technologies are planned and implemented [88]. It may be useful to use criteria to check whether a technology is socially accepted. Factors that could be used as criteria include: safety of use, privacy, comfort, simplified maintenance, weather resistance, adaptability to flooding, interest in by-products, good appearance, assurance of social status, prestige and honor [89,90].

A limitation of the sustainable and adequate technology solution in CLTS will be the economics of its implementation, as the approach is generally implemented in rural areas of low- and middle-income countries. Indeed, studies have pointed out that poverty is a key factor that contributes to households to opt for poor quality structures compared to certified and durable latrines [41,66] hence the need to find self-financing or loans mechanisms for sanitation in rural areas. However, given the high level of poverty in rural areas, the main challenge that rural households are likely to face is repaying their loans.

The costs of an improved latrine can fluctuate depending on the context, depending on variables such as the production of latrines per trained mason, the distance and terrain for a cement delivery, the distance participants travel to collect free materials for the superstructure, or the density of houses in a village. The total cost of a latrine varied from country to country, from US$27.80 in Mali to US$48.11 in Ghana [67]. But this fee may not be affordable for many households given the level of poverty in the communities where CLTS is implemented. [50] reported that households with collapsed latrine pits said they would never be able to afford a cement latrine slab on their own, as there was always a higher priority for their money, such as school fees, wedding or naming ceremonies, or home maintenance.

The low increase in the rate of access to sanitation in Sissili can be explained by the type of latrine built by the communities, most of which (87.63%) are pit latrines. According to the regulations in Burkina Faso, a household has access to sanitation if it has an improved latrine (VIP, SanPlat, flush toilet, etc.) or any other type of latrine that ensures safety for the user with a ventilation pipe, a concrete slab and limits the risk of fecal-oral contamination to a maximum. In other words, if it has a latrine in one of the "Limited", "Basic" or "Safely managed" categories of the sanitation ladder (Fig 8). The type of pit latrine that was widely used by households during the CLTS in Sissili does not meet all of these criteria and is classified as

| SERVICE LEVEL | DEFINITION |
|---|---|
| SAFELY MANAGED | Use of improved facilities that are shared with other households and where excreta are safely disposed of in situ or transported and treated offsite. |
| BASIC | Use of improved facilities that are not shared with other households. |
| LIMITED | Us of improved facilities shared between two or more households. |
| UNIMPROVED | Use of pit latrines without a slab or platform, hanging latrines or bucket latrines |
| OPEN DEFECATION | Disposal of human faeces in fields, forest, bushes, open bodies of water, beaches or other open spaces, or with solid waste |

Fig 8. SDG Sanitation Ladder [91].

"Unimproved". Households with these latrines were not considered to have access to sanitation in Burkina Faso.

Although CLTS has significantly increased latrine coverage in Sissili, it has only slightly increased the rate of access to sanitation. It therefore seems utopian that at this rate Burkina Faso will achieve the SGD6.2 ". . . Ensure access for all to adequate and equitable sanitation. . ." through CLTS unless reforms are made to the CLTS approach specifically on the technologies that it should promote.

Sanitation in general, and CLTS in particular, must be considered as an engineering activity in its own right, and therefore requires the implementation of all the necessary engineering studies (geotechnical, geological, groundwater level, etc.) to ensure that any resulting infrastructure is durable, safe and adequate. This is one of the major elements missing from the CLTS, as this technical aspect was not taken into account when the approach was designed. The emphasis was essentially on stopping open defecation by all means possible, including ensuring that households could not acquire improved latrines and be considered as having access to sanitation. The CLTS must be the subject of engineering studies to determine the choice of appropriate and safe technologies. Beyond the socio-economic aspects, the sustainability of the CLTS and the achievement of access to adequate and safe sanitation also depend on the durability and resilience of the facilities built.

## Conclusion

This study addressed one of the weaknesses of the CLTS approach, namely the short lifespan and poor quality of facilities it generates, leading to a slippage towards OD of communities that had achieved ODF status 2 to 3 years earlier. The study showed that the CLTS approach was successful overall in encouraging households to build latrines, increasing the latrine coverage rate from 29.51% to 90.44% in the Sissili province of Burkina Faso. However, 97.53% of the latrines built were unimproved pit latrines without/with slabs and superstructures made of wood or clay and no roof, many of which collapse during the rainy season. While the most resilient still rebuild their latrines, others return to the practice of OD. One of the reasons for these recurrent collapses is the lack of latrine technology guidelines and standards in the CLTS. The rate of access to sanitation in Sissili increased from 11.9% in 2016 to 17.00% in 2020 despite latrine coverage of 90.44%. This study has therefore shown that although CLTS helps to increase latrine coverage rapidly, it does not guarantee a proportional increase in the rate of access to sanitation in the community, city or region where it is implemented. This is due to the type of technologies adopted by the communities under CLTS implementation, which are not taken into account in the calculation of the sanitation access rate because they are not safe and adequate for the user and do not safely break the feco-oral contamination. In this paper, we have attempted to address these gaps in latrine sustainability by providing recommendations from a technological perspective to be incorporated into the implementation of the CLTS approach. These recommendations, if implemented, could contribute to limiting the number of collapsed latrines, ensure latrine sustainability and behavioral change and also rapidly increase the rate of access to sanitation in rural areas through CLTS. All the above intervention can contribute to the achievement of SDG6.2. Our analysis has shown that there is no single universal technology capable of solving the problem, but rather a set of different technological provisions that could be implemented taking into account the environmental, social and economic contexts in which they are inserted. It is therefore imperative to tailor solutions to each area, or even each village, and to ensure that long-term mechanisms are in place to support behavior change in a sustainable manner. The sustainability issues of CLTS and the factors surrounding them should also be a research priority in light of the spread of

the CLTS approach around the world. In addition, there is a need for further research on social approaches by combining technical and engineering aspects. This will result to improved, safe and sustainable sanitation facilities and long-term user ownership by users.

## Supporting information

**S1 File. Household survey questionnaires.**
(DOCX)

**S2 File. Interview guide.**
(DOCX)

## Acknowledgments

The authors are extremely grateful to Faith Muema her advice and input on improving the English writing style. We also thank the NGO APS and the Ministry of Water and Sanitation of Burkina Faso for the acquisition of certain data. We also acknowledge the input from the editors and anonymous reviewers who helped in improving the content and quality of this paper. And let's not forget our research institute "Institut International d'Ingénierie de l'Eau et de l'Environnement (2iE)" and data collectors.

## Author Contributions

**Conceptualization:** Hemez Ange Aurélien Kouassi.

**Data curation:** Hemez Ange Aurélien Kouassi, Harinaivo Anderson Andrianisa, Maïmouna Bologo Traoré.

**Formal analysis:** Hemez Ange Aurélien Kouassi, Harinaivo Anderson Andrianisa.

**Funding acquisition:** Harinaivo Anderson Andrianisa.

**Investigation:** Hemez Ange Aurélien Kouassi, Rikyelle Momo Nguematio.

**Methodology:** Hemez Ange Aurélien Kouassi, Seyram Kossi Sossou.

**Project administration:** Harinaivo Anderson Andrianisa.

**Resources:** Seyram Kossi Sossou.

**Supervision:** Seyram Kossi Sossou.

**Validation:** Maïmouna Bologo Traoré.

**Visualization:** Hemez Ange Aurélien Kouassi, Maïmouna Bologo Traoré.

**Writing – original draft:** Hemez Ange Aurélien Kouassi.

**Writing – review & editing:** Harinaivo Anderson Andrianisa, Seyram Kossi Sossou, Maïmouna Bologo Traoré, Rikyelle Momo Nguematio.

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
