## [Decision Letter · Decision Letter 0]

13 Jul 2023

PONE-D-23-13662Sustainability of facilities built under the Community-Led Total Sanitation (CLTS) implementation: Moving from basic to safe facilities on the sanitation ladderPLOS ONE

Dear Dr. KOUASSI,

Thank you for submitting your manuscript to PLOS ONE. After careful consideration, we feel that it has merit but does not fully meet PLOS ONE’s publication criteria as it currently stands. Therefore, we invite you to submit a revised version of the manuscript that addresses the points raised during the review process. Please see my comments below.

We look forward to receiving your revised manuscript.

Kind regards,

D. Daniel, Ph.D.

Academic Editor

PLOS ONE

Journal Requirements:

5. Please ensure that you refer to Figures 4 and 5 in your text as, if accepted, production will need this reference to link the reader to the figure.

6. We note that Figure 1 in your submission contain map images which may be copyrighted. All PLOS content is published under the Creative Commons Attribution License (CC BY 4.0), which means that the manuscript, images, and Supporting Information files will be freely available online, and any third party is permitted to access, download, copy, distribute, and use these materials in any way, even commercially, with proper attribution. For these reasons, we cannot publish previously copyrighted maps or satellite images created using proprietary data, such as Google software (Google Maps, Street View, and Earth). For more information, see our copyright guidelines: http://journals.plos.org/plosone/s/licenses-and-copyright.

Additional Editor Comments:

Dear authors,

Your submission has some merit, but needs many improvements. So I decided to give you a chance to improve it. Please see the comments from both reviewers. I have also some comments. Please note that unsatisfactory answers will result in the rejection of your draft. Good luck!

1. Please shorten your abstract to max 250 words.

2. You have very long paragraphs in the introduction, method, etc. Please use a maximum of 5-6 sentences (but also not very long) per paragraph.

3. I don’t see any knowledge gaps proposed in this study. You only mention the objective without telling what is new in your study.

4. I agree with the 2nd reviewer, your objective is not neutral. It seems that you want to prove the assumption that you describe previously. Even, I don’t see a strong argument in the previous paragraph about your assumption, i.e., the latrine technology used in CLTS doesn’t lead to improved sanitation access. Also, as a WASH practitioner, I’m a bit confused by your objective because all types of sanitation technology (Pit latrine, VIP, etc.) increase sanitation access.

5. Please organize better the section's name and numbering. You have an introduction section but then have sections 1, 2, 3 (method and tools), etc. Very messy!

6. I think the introduction section is too long. Maybe you can shorten it a bit because you also have another section about CLTS intervention in Burkina Faso. Or maybe you can put that part in the introduction, but need to shorten it a bit as well.

7. Please check your storyline. You mention Sissili in section 1, but then explain later this area in section 2.

8. Again, storyline. It is strange that you mention objectives in the method section.

9. please check the neatness of the writing, e.g., the spacing on page 7 method and tools section.

10. I think you can put the detail of the sampling calculation in the supplementary (page 8).

11. “increasing the latrine coverage rate from 29.51% in 2016 to 107.66% in 2020” -> more than 100%, also in table 2? Please explain this.

12. If you already have the graph and table, don’t need to explain in detail in the paragraph, e.g., page 14.

13. The explanation about types of latrine may not be necessary unless the definition used in the study area is different than what we know/literature say. Or you can move it to the supplementary.

14. Please edit the way you report the quotes in the journal. You must follow the “qualitative” approach here, see for example: https://www.nature.com/articles/s41598-023-30586-z

15. There are some paragraphs that consist of one sentence only. Please edit it.

16. I think the proposed guidelines may be located in the discussion section. Also, I never see any scientific journal describing step-by-step guidelines/procedures. I think you need to remove it or put it in the supplementary.

17. In the conclusion, you write “it does not guarantee a significant increase in the rate of access to sanitation”. Please remove the statistical word “significant” there.

18. You write that the main objective of your analysis is about latrine technology (in the introduction), but then your guidelines describe CLTS in general, not only latrine technology. I think you must change your main objective, otherwise focus your recommendation on the latrine technology and not general CLTS.

-- Academic Editor --

Reviewers' comments:

Reviewer's Responses to Questions

**Comments to the Author**

1. Is the manuscript technically sound, and do the data support the conclusions?

Reviewer #1: Yes

Reviewer #2: No

2. Has the statistical analysis been performed appropriately and rigorously? 

Reviewer #1: Yes

Reviewer #2: No

3. Have the authors made all data underlying the findings in their manuscript fully available?

Reviewer #1: No

Reviewer #2: No

4. Is the manuscript presented in an intelligible fashion and written in standard English?

Reviewer #1: Yes

Reviewer #2: Yes

5. Review Comments to the Author

Reviewer #1: Overall, this study provides a very insightful messeages to the global sanitation community. Indeed it has great policy and programmatic implications that need to be incorporated into sanitation policy and also programmes surrounding sanitation and community-led total sanitation. However, I hope that the could refine the manuscript, particularly the methods part. Please refer to the following comments and suggestions.

Can they rephrase their study aim, objectives and hypothesis in more neutral way? The current description regarding the study aim, objectives and hypothesis are described in a sort of biased way. For instance, they stated that “this paper seeks to provide evidence that the latrine technology used in CLTS does not increase sanitation access, and propose a few suggestions…”, and also “the objective of this study is to confirm that the latrine technology used in CLTS does not lead to sanitation access, and thus propose the ways to promote…”. As a research paper, I would suggest that the authors rephrase the study objectives in more neutral way, for instance: “we aim to assess whether the latrine technology used in CLTS did increase….in rural villages of Burkina Faso”, and I also suggest that they delete this, “and propose a few suggestions…” since this is not a policy paper and they could include their suggestions in the discussion part not in the results part.

Can the authors move all the paragraphs of 1 (intervention of the CLTS approach in Burkina Faso) to 5 (data analysis) to Methods part?

(3. Methods and tools)

The first paragraph is explaining study objectives, which should be moved to the introduction.

As for the data collected from the NGO (APS), more explanations should be added including: who conducted the surveys; when did they conduct; how much size and what methods they applied; validity and reliability of the surveys; and the like.

(sample size calculation formula)

The authors cited this “The non-response rate is considered acceptable when it is less than 10% [37]”. However, when designing sample size, it would be better to estimate the non-response rate in a conservative manner so that they could sample as large as possible. I guess that 5% of non-response is too small. I would suggest that they describe somewhere in the results part about the actual non-response rate in their survey.

The WHO manual for sample size estimation explains different formula for different situations and it would be better if authors could explain the rationale why they used the particular formula. In addition, I hope they could rephrase the explanations about the P in the formula. I guess that P in the formula could be redescribed as latrine coverage instead of maintaining current description. If that was true, they also have to explain why they assumed that the latrine coverage was 50% (meaning P=0.5). In fact, they explained that 7 villages were selected in page 8, which means that they seemed to have applied “two-stage sampling methods”; however, the sampling formula they used were based on simple random sampling.

Can they explain how those 7 villages were selected and what particular sampling methods were used? Based on their explanation, they seemed to have used purposive sampling. If that was the case, next question is why they decided to sample “seven (7)” villages, which is related to sample size calculation. In page 9, they stated that they used “the probability proportionate to size” methods, which is called PPS sampling. The PPS sampling is used when selecting clusters (which means village in this paper) not households or individuals. In the same paragaph (the first paragraph in page 9), they explained how they calculated the number of households and this are not related to the PPS sampling. Normally for the PPS sampling, the sample size per village (cluster) is same regardless of the village population because the probability of selecting a village of larger population became higher than the smaller villages. I guess they allocated sample size per village in proportionate to the village population. I hope they could rephrase all the paragraphs related to sampling, and explain the limitations of the sampling methods somewhere in the discussion part.

Please add the table number in page 8, and also “,” should be changed to “.”. Alternatively, I would suggest that they delete this table since it is commonly accepted table in basic statistics, and all the values in the table were not developed by the authors.

“Oral informed consent…”.

This sentence should be moved to the part of Ethical considerations.

Can they provide the questionnaire set as a supplementary material?

“The household surveys were conducted during the rainy season in two campaigns.”

Does this mean that they conducted the one-time survey for two years (2021 and 2022)? Or did they conduct two rounds of survey targeting 410 households? If the first was true, how could they explain the one year difference between the two surveys? Why did the need 2 years for conducting relatively small scale survey?

They should describe in more detail regarding the respondents (stakeholders) other than households in the Table 1 since most of the explanations in page 8-10 were only about the households sampling. For example, why and who interviewed the NGO implementers, institutional/state actors, community stakeholders. Also, please provide interview guide or questionnaire (even it happened to be unstructured) as a supplementary material.

4. ethical consideration

Can they provide more explanations about the ethical approval from the ethical committee or ethical review board including the approval number?

5. data analysis

“as recommended by [41]”

Can they repharase this by adding the scholars’ names to the sentence instead of just putting reference number?

Having read all the manuscript, I think theis study is quantitative rather than qualitative study. The majority of the results are based on quantitative analysis. I would suggest that the authors either delete explanations about qualitative methods and their results or add more explanations on methods and the results about the qualitative analysis both in methods and results part.

6. Result

Introduction, Methods, Results and Discussion are the same level of heading, and thus number 6 is not appropriate for the Results part.

The paragraph includes explanations on “methods”. Can they explain some part of this paragraph also in the methods while maintaining this paragraph in the results as they are?

I would suggest that they maintain the Table 3 in the main body of the manuscript and attach the Table 2 as a supplementary material for simplicity instead of presenting the two tables in the main body.

Can they explain more details about the Table 2 and 3 in methods? I would expect that the authors will describe details about the 7 surveyed villages, but they explained much details about the coverage in other villages in the Results, also the number of villages was also 7. I hope they could exlain more about the methods and data they got from APS & INSD in the methods part. In additin, add some explanations as a footnote for the Table 3.

In table 5, Can they add % in addition to the number?

In page 23, can they delete “the sad fact” since it is subjective expression.

The all paragraphs starting from the “This low….” in page 24 should be moved to the discussion part since it is not results.

Discussion

In the first paragraph of the discussion in page 26, “to some information” should be deleted or rephrased.

Can they include the limitations of this study in the discussion part?

Reviewer #2: This manuscript offers important reinforcing evidence of the shortcomings of CLTS that have been raised elsewhere but which it does not cite (see for example: Trimmer, J et al. 2022. The Impact of Pro-Poor Sanitation Subsidies in Open Defecation-Free Communities: A Randomized, Controlled Trial in Rural Ghana. Environmental Health Perspectives. DOI:10.1289/EHP10443; Brown, J., et al. 2019. Community-Led Total Sanitation Moves the Needle on Ending Open Defecation in Zambia. American Journal of Tropical Medicine and Hygiene 100 (4) 767-769. DOI:10.4269/ajtmh.19-0151; and Cameron, L., et al. 2021 Sanitation, financial incentives and health spillovers: A cluster randomised trial. Journal of Health Economics. DOI:10.1016/j.jhealeco.2021.102456). Its interviews with households over two seasons along with field observations of the sanitation infrastructure are important, particularly as they lend themselves to the sobering recognition of the durability problem as perhaps the most critical shortcoming of CLTS. The authors’ questions about the effectiveness of CLTS’ zero-subsidy principles are also critical to understanding CLTS’ slippage issues.

In its current state, however, this manuscript is unpublishable in a peer-reviewed venue. Most problematically, its stated objectives are inconsistent with the core scientific convention of posing neutral questions which are either answered (or not) by the evidence collected (see page 7 of the manuscript). Determining that

“CLTS increased coverage,” or that “the quality of latrines built under CLTS is poor” are statements of results, not neutral research objectives; indeed, framed as research objectives, they reveal bias. Instead, the objectives should be framed as questions, e.g. “how did CLTS change sanitation coverage in Sissili Province?” or “what is the failure rate of latrines installed under the CLTS program in Sissili Province?” or “what explains the high failure rate of latrines installed under CLTS in Sissili Province?”

There are other significant issues. For example, the authors report two waves of data collection, but do not report results from each wave, nor do they do indicate how the two waves of data collection differed. From what I could tell, they report the two waves of household surveys as though it were a single cross-sectional survey collected during a single survey period, but it’s not clear.

Also: much of the results section presents data unrelated to the authors’ own household survey results. Tables 2 and 3 and Figure 8 all correspond to “corrected” self-reported latrine coverage data from the NGO APS.

Meanwhile, the objectives of the power calculations for the household surveys are not obvious to me. Typically power calcs are designed to communicate the smallest detectable difference between an intervention group and a counter-factual group with respect to a specific variable of interest in a controlled trial, or else, in the case of a quasi-experimental study, to explain the variation in a response variable with respect to the variation of one or more explanatory variables, or to determine the sample’s “representativeness” of a broader population. None of that is offered here. At best, this is an effort to collect some “ground truth” on CLTS performance in Sissily Province that may be illustrative of broader trends but not conclusive.

Also: the authors make claims about how CLTS has affected open defecation in Sissily province (they use the term “cessation of OD”) but they neither measure OD themselves nor refer to any other studies of open defecation (as distinct from latrine coverage or “access.”)

Also: the recommendations regarding CLTS are intriguing, but they don’t necessarily follow directly from the data presented or directly collected by the authors, and are better suited for some kind of commentary or policy analysis piece, rather than a research article as presented here.

In short, I very much hope that a sensible venue and format can be found for the team’s findings of latrine durability challenges and its associated drivers related to affordability of durable materials and professional latrine installation, but I don’t see an academic journal research article as offering them.

6. PLOS authors have the option to publish the peer review history of their article (what does this mean?). If published, this will include your full peer review and any attached files.

Reviewer #1: No

Reviewer #2: No

---

## [Author Response · Author response to Decision Letter 0]

16 Aug 2023

Responses to comments from Reviews and Editor have been attached in a separate file entitled "Response to Reviewers and Editor", which we have included in this submission of our revised manuscript.

---

## [Decision Letter · Decision Letter 1]

12 Oct 2023

Sustainability of facilities built under the Community-Led Total Sanitation (CLTS) implementation: Moving from basic to safe facilities on the sanitation ladder

PONE-D-23-13662R1

Dear Dr. KOUASSI,

We’re pleased to inform you that your manuscript has been judged scientifically suitable for publication and will be formally accepted for publication once it meets all outstanding technical requirements.

Kind regards,

D. Daniel, Ph.D.

Academic Editor

PLOS ONE

Additional Editor Comments (optional):

Reviewers' comments:

Reviewer's Responses to Questions

**Comments to the Author**

1. If the authors have adequately addressed your comments raised in a previous round of review and you feel that this manuscript is now acceptable for publication, you may indicate that here to bypass the “Comments to the Author” section, enter your conflict of interest statement in the “Confidential to Editor” section, and submit your "Accept" recommendation.

Reviewer #1: All comments have been addressed

2. Is the manuscript technically sound, and do the data support the conclusions?

Reviewer #1: Yes

3. Has the statistical analysis been performed appropriately and rigorously? 

Reviewer #1: Yes

4. Have the authors made all data underlying the findings in their manuscript fully available?

Reviewer #1: Yes

5. Is the manuscript presented in an intelligible fashion and written in standard English?

Reviewer #1: Yes

6. Review Comments to the Author

Reviewer #1: All the comments in my review were adequately addressed in this revised version and I feel that this manuscript is now acceptable for publication

7. PLOS authors have the option to publish the peer review history of their article (what does this mean?). If published, this will include your full peer review and any attached files.

Reviewer #1: No

---

## [Editor Report · Acceptance letter]

8 Nov 2023

PONE-D-23-13662R1 

Sustainability of facilities built under the Community-Led Total Sanitation (CLTS) implementation: Moving from basic to safe facilities on the sanitation ladder 

Dear Dr. Kouassi:

I'm pleased to inform you that your manuscript has been deemed suitable for publication in PLOS ONE. Congratulations! Your manuscript is now with our production department. 

Kind regards, 

on behalf of

Mr D. Daniel 

Academic Editor

PLOS ONE